# Paracrine Regulation and Immune System Pathways in the Inflammatory Tumor Microenvironment of Lung Cancer: Insights into Oncogenesis and Immunotherapeutic Strategies

**DOI:** 10.3390/cancers16061113

**Published:** 2024-03-10

**Authors:** Firas Batrash, Adnan Shaik, Rayaan Rauf, Mahmoud Kutmah, Jun Zhang

**Affiliations:** 1School of Medicine, University of Missouri-Kansas City, Kansas City, MO 64108, USA; 2Division of Medical Oncology, Department of Internal Medicine, University of Kansas Medical Center, Kansas City, KS 66160, USA; 3Department of Cancer Biology, University of Kansas Medical Center, Kansas City, KS 66160, USA

**Keywords:** lung cancer, paracrine regulation, immunotherapy, TME

## Abstract

**Simple Summary:**

Despite massive strides taken across the board in oncology, there remain gaps in understanding the relationship between cancer cells and the body’s immune system, tissues, and signaling pathways. This review explores some of the recent steps made toward filling these gaps and understanding how certain cytokine signals create a tumor microenvironment that facilitates the growth and survival of cancer cells. New treatment approaches targeting these facilitators have been developed as potential disruptors of tumor growth. Lastly, a discussion of current gaps in the research can help navigate new directions to continue life-saving research for lung cancer treatments.

**Abstract:**

The intricate interplay between inflammatory processes and the tumor microenvironment (TME) in lung cancer has garnered increasing attention due to its implications for both oncogenesis and therapeutic strategies. In this review, we explore recent advances in understanding the paracrine regulation and immune system pathways within the inflammatory TME of lung cancer. We delve into the molecular mechanisms underpinning oncogenesis, highlighting the role of immune cell populations, cancer-associated fibroblasts, and endothelial cells, as well as their interactions through immune system pathways regulated in a paracrine pattern. Additionally, we discuss emerging immunotherapeutic strategies with a specific focus on the potential of leveraging the inflammatory TME through these pathways to enhance treatment efficacy in lung cancer.

## 1. Introduction

Inflammation, a crucial component of the immune response, has dual roles in lung cancer—fighting acute infections and contributing to chronic conditions conducive to carcinogenesis. The tumor microenvironment (TME) encompasses immune cells, cancer-associated fibroblasts (CAFs), cancer cells, and endothelial cells that orchestrate paracrine signaling through growth factors, cytokines, and chemokines. Understanding the intricate interplay between inflammation, the immune response, and the TME is essential for unraveling the complexities of lung cancer development and devising effective therapeutic interventions.

## 2. Paracrine Regulation and Immune System Pathways

### 2.1. Paracrine Regulation in TME

Immune cells, including tumor-infiltrating lymphocytes (TILs), macrophages, dendritic cells (DCs), and regulatory T cells (Tregs), etc. play a central role in the lung cancer inflammatory TME [1,2,3,4]. These cells secrete a diverse array of cytokines, chemokines, and growth factors that exert potent paracrine effects on neighboring cells [3,5,6]. For instance, proinflammatory cytokines such as interleukin-6 (IL-6) and tumor necrosis factor-alpha (TNF-α), released by T lymphocytes and activated macrophages, can stimulate cancer cells and stromal cells to produce other factors that enhance tumor survival, proliferation, and angiogenesis [5,6,7]. Conversely, immune cells can be influenced by cancer cells to adopt an immunosuppressive phenotype through paracrine communication, fostering an environment that hampers effective antitumor immune responses [6].

Cancer-associated fibroblasts (CAFs), critical components of the stromal compartment in the TME, actively participate in paracrine regulation [8,9,10]. CAFs are known to secrete growth factors like fibroblast growth factor (FGF) and transforming growth factor-beta (TGF-β), which can stimulate cancer cell proliferation, epithelial–mesenchymal transition (EMT), and invasiveness [8,9,10,11]. Moreover, C-X-C-type chemokines including CXCL8 (IL-8), CXCL2, and CXCL12 secreted by CAFs are also involved in tumor proliferation by promoting angiogenesis and metastasis (CXCL8 & CXCL12) and increasing PD-L1 expression (CXCL2) [9,12,13,14]. Additionally, CAFs contribute to extracellular matrix remodeling, creating a supportive niche for tumor growth [9,15]. Paracrine interactions between CAFs and immune cells can influence the immunosuppressive milieu by promoting the recruitment of immunosuppressive cells like Tregs and myeloid-derived suppressor cells (MDSCs) [9]. Additionally, CAFs have been noted to directly diminish the antitumor activities of natural killer (NK) cells, suppress the activity of T cells through the chemokine-mediated stimulation of a checkpoint regulator expression on cancer cells, and promote the M2 polarization of macrophages in the TME [9].

The development of new blood vessels (angiogenesis) is crucial for supplying nutrients and oxygen to growing tumors [1]. Endothelial cells within the TME engage in paracrine signaling with cancer cells and other stromal components [1]. Factors such as vascular endothelial growth factor (VEGF) released by cancer cells stimulate endothelial cells to sprout and form new blood vessels [1,5]. Additionally, VEGF suppresses T cell activity, increases the recruitment of Tregs and MDSCs, and inhibits DCs, contributing to antitumor immunity [16]. This process is a hallmark of tumor progression and can be influenced by immune cells as well, further highlighting the complexity of paracrine interactions in the inflammatory TME [1].

### 2.2. Immune System Pathways

The immune system’s intricate involvement in the TME of lung cancer is a dynamic interplay that shapes the tumor’s fate, progression, and response to therapy. Immune cell populations, cytokines, chemokines, and immune checkpoints collaborate in a complex network of pathways that dictate whether the tumor is eliminated, contained, or allowed to thrive [3,17].

Tumor-infiltrating lymphocytes (TILs), including cytotoxic CD8+ T cells and helper CD4+ T cells, form a frontline defense against cancer cells in the TME [18]. These immune cells recognize tumor antigens and initiate immune responses that aim to eliminate malignant cells [19,20]. The activation of TILs occurs through antigen presentation and subsequent immune signaling pathways [19]. T cell receptor (TCR) engagement with major histocompatibility complexes (MHCs) on cancer cells activates signaling cascades, including the phosphoinositide 3-kinase (PI3K)-AKT-mammalian target of rapamycin (mTOR) pathway, leading to T cell proliferation, cytokine secretion, and cytotoxicity [19,21].

Immune checkpoint molecules (ICMs), including molecules like PD-1 and cytotoxic T-lymphocyte associated protein (CTLA)-4, are crucial regulators of immune responses that maintain self-tolerance and prevent excessive immune activation [22,23]. However, tumors exploit these checkpoints to evade immune surveillance [23,24]. For example, in lung cancer, programmed cell death protein 1 (PD-1) on T cells interacts with its ligand PD-L1 on cancer cells, inducing T cell exhaustion and immune suppression [22,25]. Blocking this interaction, or others like it, using checkpoint inhibitors unleashes T cell responses against cancer cells [25].

The tumor-associated macrophages (TAMs), a type of immune cell, are significant contributors to the inflammatory milieu within the TME. These cells can adopt either a proinflammatory (M1) or anti-inflammatory (M2) phenotype, influenced by pathways such as nuclear factor-kappa B (NF-κB) and JAK-STAT3 [26,27]. M1 TAMs secrete cytokines like interleukin-12 (IL-12) that promote antitumor responses, while M2 TAMs secrete immunosuppressive factors that aid tumor progression and metastasis through various functions [27,28]. Other factors closely linked with inflammation in lung cancer include IL-1β, IL-4, IL-8, IL-11, IL-12, monocyte chemotactic protein (MCP)-1, and TGF-β [3,27,28]. The balance between M1 and M2 polarization is essential in shaping the TME’s immune landscape [27,29]. Notably, through cytokines like IL-4, the TME polarizes more toward M2 phenotypes, which demonstrate lower levels of IL-12 and IL-23 with more IL-10, aiding tumor progression by promoting angiogenesis, EMT, and metastasis [3,27,29].

Chemokines guide immune cell trafficking to specific locations within the TME, promote EMT, and influence the distribution and behavior of immune cells [28,29]. Specifically, chemokines are involved in the recruitment of tumor-associated neutrophils (TANs) to the TME [30]. The polarization from the more antitumor N1 phenotype to the protumor N2 is likely mediated by cancer and immune cell cytokines like TGF-β [30,31]. Moreover, they are involved in this complex crosstalk between the tumor stroma and cancer cells that make up the TME [28].

DCs represent one of the principal antigen-presenting cells in the immune system, typically involved in adaptive immunity T cell activation. However, in an inflammatory TME rich in IL-6, the subsequent activation of STAT3 pathways downregulates DC maturation. Other paracrine signals like IL-10 and VEGF also show inhibitory effects on DC function [32]. Like Tregs, MDSCs represent another cell responsible for immunosuppression in the TME through their reduction in CD8+ T cells and promotion of T cell apoptosis through their production of TGF-β [33]. Additionally, they play a role in suppressing the immune response against invading tumor cells, promoting EMT and stimulating angiogenesis, creating an environment susceptible to metastasis [33,34,35]. NK cells represent the opposite end of the immune spectrum, more focused on innate immunity-mediated tumor cell death through the release of cytolytic enzymes or the expression of the apoptosis-promoting Fas ligand. However, metastasis-initiating cancer cells can auto-induce quiescence through Dickkopf WNT signaling pathway inhibitor (DKK)-1. This dormancy allows these metastasizing cancer cells to evade NK-cell killing and re-establish a growth state afterward [36].

## 3. Molecular Mechanisms of Oncogenesis

Exploring the interactions within the TME is critical to creating a thorough understanding of the tumor growth cycle needed in the current age of oncology. Exploring each cell in isolation helps identify unique or common pathways and molecules that they utilize to affect the cells around them and, ultimately, contribute to carcinogenesis. Various mechanisms that result in the chronic inflammation, hypoxia, EMT, immunosuppression, angiogenesis, proliferation, migration, and invasion of cancer cells are important to the understanding of carcinogenesis that goes beyond a unilateral image that focuses solely on genetic mutations. Moreover, understanding the involvement of paracrine signaling on the primary killer of tumor cells, the immune system, is crucial for identifying the gaps that allow cancer cells to grow unregulated and unchecked. 

Chronic inflammation in the context of the TME of lung cancer is a dynamic process that contributes to genetic and epigenetic alterations in cancer cells, thereby facilitating carcinogenesis. Chronic inflammation creates an environment rich in proinflammatory cytokines and chemokines, which can enhance tumor progression and have direct effects on promoting cancer cell growth and survival (Table 1). 

### 3.1. COX-2/PGE2

Cyclooxygenase-2 (COX-2) is an enzyme responsible for the formation of prostaglandins (PGs) and has been found to be a factor of proliferation and survival in lung cancer and can even be used as a marker for metastasis [37]. Uniquely, COX-2 was found to be expressed at elevated levels in various lung cancer types and precursors and specifically in NSCLC [37,38]. Understanding the correlation between COX-2 expression and cancer is multifaceted as it encompasses various mechanisms of survival: apoptosis inhibition, angiogenesis, immunosuppression, EMT, and invasion [37,39]. An elevated COX-2 expression is linked with inflammation, IL-1β, TGF-β, EGF, mutant KRAS or TP53, hypoxia, the loss of IL-10 receptor expression, and constitutively localized STAT-6 [40].

PGE2 and its prostanoid receptor EP4 have specifically been identified to increase the matrix metalloproteinase (MMP)-2 and CD44 expressions and subsequent NSCLC tumor invasion and angiogenesis [39]. These effects were found both in autocrine and paracrine forms as the overexpression of COX-2 enzymes consequentially results in increased tumor cell EP4 expression [39]. Another mechanism of PGE2-mediated invasion in NSCLC includes the suppression of the E-cadherin expression through the induction of the transcriptional repressors ZEB1 and Snail, thereby reducing intercellular adhesion and enhancing tumor invasion and metastasis, contributing to its poor prognostic indication [41].

COX-2 and PGE2 upregulation are also involved in another important mechanism of survival: immunosuppression [42,43]. Primarily through the increase in FOXP3 in both CD4+CD25+ (Treg) and CD4+ CD25- T lymphocytes that causes the latter to switch into a regulatory phenotype, PGE2 at high levels stimulates an immunosuppressive shift in the TME [42,44]. Moreover, COX-2 is involved in the suppression of DC function as well, with a reduction in various surface molecule expressions that hinder their abilities to present antigens, induce alloreactivity, or secrete IL-12 [45].

COX-2 plays a role in increasing the apoptotic threshold of tumor cells in NSCLC through its stabilization and subsequently reduced ubiquitination of the molecule survivin, a spindle microtubule binding protein involved in apoptotic avoidance independent of the cell cycle [46]. Moreover, COX-2 and PGE2 play roles in increasing the expressions of chemokines CXCL8 (IL-8) and CXCL5 via nuclear translocation NF-κB [43]. Through the upregulation of this process, COX-2 has been found to increase tumor growth and angiogenesis in vivo [43]. 

### 3.2. TGF-β

Transforming growth factor (TGF-β) is part of a polypeptide family including 3 TGF-β isoforms, Nodal, bone morphogenetic proteins (BMPs), activins, and a few others found to be impaired in many cancer types including lung cancer, where it has antiproliferative functions [47]. TGF-β is secreted from most cells in the TME, including fibroblasts, immune cells, and epithelial cells. Functioning through two transmembrane receptors, TGFRI and TGFRII, TGF-β induces downstream signal cascades through Smad2 and Smad3, which then phosphorylates Smad4, which translocates to the nucleus [48]. This process undergoes feedback inhibition through Smad6 and Smad7, which recruit Smurf to disrupt the signaling at the receptor level [49]. Non-canonically, the transduction may utilize NF-κB, PI3K, or MAPK, making the signaling stream complex and widespread in effect [50,51]. TGF-β has been identified to play two roles in tumors, one of tumor suppression early on and later, a more tumorigenic role in mediating EMT and other aspects of the TME [48,50,51]. This EMT is also potentially subject to crosstalk with TNF-α, which enhances the EMT by causing cancer cells to switch to a cytokine- and chemokine-secreting phenotype, increasing the capacity for invasion [52]. Moreover, CAFs also play an integral role in this process through their secretion of IL-6 under TGF-β-mediated fibrosis [53,54].

The cytostatic function of TGF-β is mediated by its induction of the CDK inhibitor (i.e., p15, p21) expression, reduction in c-Myc expression, and several other mechanisms downstream of Smad, which together help arrest the cell cycle in the G1 phase. In terms of its protumor function, TGF-β plays an important role in the function of TANs, inducing the immunosuppressive N2 polarization, which inhibits NK cells, recruiting Tregs and anti-inflammatory M2 macrophages, and releasing angiogenic MMP9 [30,31]. In the absence of this signal, the N1 polarization can target tumor cells directly or through their influence on other proinflammatory immune cells [31]. Moreover, more recent research has focused on its role in the protumor metabolic transitions of the TME including CAFs, immune cells, and cancer cells themselves. For example, by promoting “reverse Warburg effects” in CAFs, whereby, a high rate of aerobic glycolysis utilized to transport metabolites to cancer cells demonstrated a key function of TGF-β and a potential explanation for the survival of cancer cells during metastasis before angiogenesis can take place [51,55]. 

BMPs, members of the same family, have also been identified to be involved in the tumorigenic process. Namely, BMP-2, BMP-4, and BMP-7 are correlated with poor prognosis in lung cancer through their induction of angiogenesis and tumor growth [56].

### 3.3. EGF

Epidermal growth factor receptor (EGFR)-mediated activations of MAPK, PI3K, and STAT3 signaling pathways demonstrate important mechanisms by which various ligands (i.e., EGF, TGF-α, amphiregulin, etc.) influence tumor growth, survival, and metastasis [57,58,59]. Notably, upregulated EGF and amphiregulin are associated with poor prognosis in NSCLC including malignant metastasis and resistance to targeted inhibitors as part of an intricate autocrine EGFR growth loop [57,60,61]. In terms of apoptotic evasion in NSCLC, amphiregulin inactivates the proapoptotic BAX, while the inhibition of EGFR expression and activity see an upregulation in the proapoptotic markers and decreases in the tumorigenic markers [60,62,63]. Specifically, EGF has an ability to stimulate the production of angiogenic factors like VEGF, bFGF, and hypoxia-induced factor 1 (HIF-1) downstream of its MAPK or PI3K cascades [56].

Moreover, EGF’s involvement in tumorigenesis extends to tumor migration as well. While the focus is generally on TGF-β mediated Smad signaling, there is some involvement of EGF-mediated MAPK signaling to instigate migration; however, in A549 lung adenocarcinoma cells, EGF had no effect on the EMT marker’s MMP2 expression, and EGF was not responsible for significantly increasing the invasive potential of these cells, highlighting the importance of TGF-β to EMT and invasion as discussed [64]. However, EGFR may participate in immune evasion in NSCLC through a potential regulation of the B7-H5 expression [62]. Additionally, the activation of this pathway is involved in inducing PD-L1 expression as EGFR can stimulate the IL-6/JAK/STAT3 pathway in NSCLC cells [65].

### 3.4. FGF

Another important growth factor to be discussed in its role in lung cancer is fibroblast growth factor (FGF). One such member of this family specifically identified in high levels in NSCLC tumors is the basic fibroblast growth factor (bFGF/FGF2) [66]. Utilizing its receptor FGFR, FGF signals to several key downstream pathways, including MAPK, PI3K, PLC, and STAT, promoting several tumorigenic mechanisms. While several mechanisms of the involvement of FGF exist across different cells of the TME, in lung cancer, FGF may play a potential role in inducing IFN-γ-mediated PD-L1 expression, diminished T cell infiltration, Treg generation, and T cell depletion [67]. Moreover, the upregulation in FGFR signaling plays an important role in tumor growth by promoting the survival of MDSCs, recruitment of TAMs, EMT, and angiogenesis [67,68]. The role of FGF is particularly important in the understanding of CAFs in the TME, which overexpresses FGF9 and bFGF and secrete bFGF in significant amounts that contribute to tumor cell growth in lung adenocarcinoma cell models [68]. FGF may also play a role in angiogenesis as an upstream stimulator of VEGF and PDGF as well as a regulator of newly formed vessels whereby they have the potential to activate Bcl-2 and inhibit apoptosis in endothelial cells or MAPKs [69].

### 3.5. Factors of Angiogenesis: VEGFA, HIF-1α, CSF, and PDGF

Angiogenesis is a crucial process for tumor cell growth, nutrient acquisition, metastasis promotion, and inflammation and is thus an important aspect to understand [56]. Several stromal cells are deeply involved in this process, including macrophages, which secrete angiogenic growth factors and degrade the perivascular ECM, neutrophils, which are important for triggering angiogenesis, lymphocytes, which can be anti- or pro-angiogenic, and CAFs, which produce the new ECM and more angiogenic factors. Overall, the TME may also take on different metabolic properties in order to provide energy to fuel angiogenesis [70].

One important mechanism for this process in lung cancer utilizes overexpressed vascular endothelial growth factor (VEGF) to drive the formation of blood vessels in and around the tumor [56]. The main angiogenic VEGF, VEGF-A, utilizes two tyrosine kinase receptors, VEGFR-1 and VEGFR-2, and through their signaling, VEGF can stimulate angiogenic signals in endothelial cells as well as the secretion of von Willebrand factor via VEGFR-2 and a weaker pro-angiogenic activity from VEGFR-1 [56,71]. VEGFR-3 and VEGF-C/D may also be involved in tumor angiogenesis [71]. Like VEGF, platelet-derived growth factor (PDGF) receptors have also been found to be overexpressed in lung cancer cells and linked to a poor prognosis [56]. The regulation of VEGF through other mechanisms has also been explored in NSCLC as p53 and Bcl2 were significantly associated with VEGF expression; however, a recent study demonstrated no difference in survival regarding the Bcl2 and VEGF statuses for patients with advanced NSCLC despite the identified correlation [72,73].

As the solid tumor expands, the hypoxic conditions, due to a lack of blood supply, promote cancer cell survival and angiogenesis by activating HIF-1α, inducing anaerobic glycolysis, inhibiting apoptosis, and stimulating the expressions of pro-angiogenic factors like VEGF. This alters the TME to support tumor growth and immune evasion [74]. Notably, elevated levels of HIF-1α, VEGF, and CCL28 are linked to a greater infiltration of Treg cells in lung adenocarcinoma samples, and high levels of HIF-1α and HIF-2α are associated with poor prognoses in both SCLC and NSCLC [75,76]. Additionally, hypoxia’s impact on immune chemokine secretion in SCLC tumor cells is known to attract immunosuppressive cells [75]. 

Another group of cytokines, colony-stimulating factors (CSFs), which typically focus their effects on white blood cell proliferation and differentiation, also aid in angiogenesis. The four main types, G-CSF, GM-CSF, M-CSF, and IL-3, function on different cell types, and each may play a role in lung cancer angiogenesis. G-CSF and GM-CSF are particularly associated with aggressive and angiogenic lung tumors [56,77]. Other direct effects of G-CSF include promoting survival, proliferation, and migration in cancer cells, stimulating M2 polarization, and increasing MDSC and Treg phenotypes [77]. Interestingly, G-CSF has also been found to promote metastasis through its mobilization of Ly6G+Ly6C+ granulocytes to other sites prior to tumor cell arrival. These granulocytes produce Bv8 protein, which may contribute to premetastatic angiogenesis [78].

### 3.6. Other Cytokines

Under the influence of chronic inflammation, proinflammatory cytokines and chemokines such as IL-6, IL-17, and IL-23 secreted by immune cells stimulate fibroblasts within the TME, promoting cancer cell growth and survival [79]. These CAFs can then actively participate in paracrine signaling. Fibroblasts secrete growth factors, such as FGF and TGF-β [80]. These growth factors have the capacity to then stimulate cancer cell proliferation, promote EMT, and enhance tumor invasiveness as discussed. Further, the presence of integrin α11β1, a receptor for collagen XIII on CAFs, governs ECM stiffness, IGF-2 secretion, and, consequently, metastasis and NSCLC tumor growth. Integrin α11β1 influences lysyl oxidase-like 1 (LOXL1), an ECM cross-linking enzyme critical for tumor growth and invasion [81]. 

Various other cytokines exist that, depending on their secretion and target cells, can produce varying cancer-promoting effects. One such cytokine is IL-1β, a ligand for the IL-1R that produces in vivo and in vitro carcinogenic changes downstream of MAPK and NF-κB that promotes immunosuppression, angiogenesis, and the invasion of cancer [3,82,83]. Mainly expressed in innate immune cells (i.e., monocytes, macrophages, and DCs), IL-1β demonstrates a significant facilitator in lung cancer metastasis and growth through these paracrine interactions [3,83]. For example, IL-1β promotes EMT in cancer cells, PG production and leukocyte adhesion on endothelial cells, proteinase production and COX-2-mediated angiogenesis in stromal cells, immunosuppression through induced MDSCs, as well as protumor cytokine production, like that of IL-22 [82,84]. Additionally, IL-1β can stimulate VEGF, COX-2, and TGF-β to drive tumorigenesis through the angiogenic, apoptosis-resistant, and immunosuppressive mechanisms discussed for each of these factors [82]. However, IL-1β upregulation may have the additional effect of inducing mutations in TP53 that are associated with NSCLC and elevated IL-1β expressions [82,85]. 

IL-4, released by Th2 lymphocytes, is involved in regulating the immune response and has been found in high expressions in lung cancer [86]. IL-8, also known as CXCL8, is another proinflammatory cytokine that dominates inflammation signaling via NF-κB whereby it recruits MDSCs and immunosuppressive neutrophils and can also stimulate EMT and promote angiogenesis [86]. Finally, TNF-α is a key activator of NF-κB, and its effects on apoptotic evasion and cell survival will be discussed below.

Interferons are cytokines that can be divided into three types: type 1, which includes IFN-α, IFN-β, IFN-ε, IFN-κ, IFN-τ, and IFN-ω; type 2, which is IFN-γ; and type 3, which contains IFN-λ1, IFN-λ2, IFN-λ3, and IFN-λ4 [87]. Type 2, IFN-γ, has been found to be associated with tumor progression. Antigen-specific T cells that accumulate in the TME cause an increase in the IFN-γ concentration in the tissue. The IFN-γ can cause an increased expression of PD-L1 in lymphatic endothelium, which further limits cytotoxic T cell access to the TME [88]. IFN-γ is also associated with IDO upregulation. IDO is an enzyme that can suppress T and NK cells and is proposed to be freed up by an IFN-γ-stimulated macrophage degradation in tryptophan, allowing it to inhibit T and NK cells [89,90].
cancers-16-01113-t001_Table 1Table 1Summary of cytokine releases based on cell type in lung cancer TME [1,3,5,6,7,8,9,10,11,12,13,14,15,26,27,28,29,30,31,37,38,39,40,41,42,43,44,45,46,48,49,50,51,52,53,54,55,56,57,58,59,61,62,63,64,65,66,67,68,69,70,71,75,77,78,80,82,83,84,85,86,91,92,93,94,95,96,97,98,99,100,101].CellCytokine ReleasedEffectCAFFGFPD-L1 expressionTAM recruitmentTreg generationT cell depletionMDSC survival EMTVEGF- and PDGF-mediated angiogenesisApoptosis regulation in angiogenesis via bcl-2TGF-βEMT Angiogenesis via MMP9M2 and N2 polarization of TAM and TANTreg recruitmentNK cell inhibitionPGE2ZEB1-, Snail-, and MMP2-mediated ECM dysfunction immunosuppression of DCsStabilization of TregIL-6STAT3-mediated cancer cell proliferation and invasionVEGFAngiogenesisTAM (M2)IL-1βEMTInvasionVEGF- and COX-2-mediated angiogenesisMDSC-induced immunosuppressionTGF-β stimulationInduction of TP53 mutationsTGF-βEMT Angiogenesis via MMP9M2 and N2 polarization of other TAM and TANTreg recruitmentIL-10ImmunosuppressionIL-6Stimulation of pro-cancer fibroblast signalingAngiogenesisProliferationImmunosuppressionApoptotic evasionEMTInvasionMetastasisTNF-αChronic inflammationNF-κB-mediated tumor cell proliferationApoptotic evasionRelease of angiogenic factorsTAN (N2)MMP9VEGFIL-8Production of pro-angiogenic factorsTregIL-10ImmunosuppressionTGF-βTreg recruitmentMDSCIL-10ImmunosuppressionCancer CellsPGE2ZEB1-, Snail-, and MMP2-mediated ECM dysfunction immunosuppression of DCsStabilization of TregTGF-β“reverse Warburg effects” in CAFEMT Aangiogenesis via MMP9M2 and N2 polarization of TAM and TANTreg recruitmentNK cell inhibitionEGFApoptotic evasion via survivin, bcl-2, and BAXProduction of angiogenic factorsFGFTAM recruitmentMDSC survivalEMTAngiogenesisT cell depletionTreg generationCSFWBC proliferationAngiogenesisIncreased tumor aggressionM2 polarizationIncreasing MDSC and Treg phenotypesMetastasis via Ly6G+Ly6C+ granulocyte mobilizationVEGFAangiogenesisIL-1βProtumorigenic signaling in CAFTAN recruitmentAngiogenesisInduction of MDSCLeukocyte adhesion on endothelial cellsIL-22 productionIL-4Regulation of immune responseIL-6Treg and MDSC upregulationImmunosuppressive modulation of NK, neutrophil, and T cell activitySuppression of apoptosisStimulation of CAF growth factor releaseIL-8Recruitment of MDSC and immunosuppressive neutrophilsEMTAngiogenesisIL-12/23TAN recruitment

### 3.7. Key Intracellular Signals

One of the key transcription factors activated by chronic inflammation in the TME is NF-κB. The activation of this factor can involve two different pathways: canonical and non-canonical. The canonical pathway of NF-κB is active during responses involving inflammation and immune responses. It is particularly critical in modulating innate immunity [102]. On the other hand, the non-canonical pathway for activating NF-κB is required for lymphoid organ development as well as adaptive immunity [103]. The canonical pathway is initiated by triggers like proinflammatory cytokines TNF-α and IL-1, as well as bacterial substances such as lipopolysaccharide (LPS). These triggers activate IκB kinases (IKKs), which subsequently phosphorylate the primary inhibitor of NF-κB: IκBα. This phosphorylation step results in the ubiquitination and subsequent degradation in IκBα by the proteasome. The NF-κB complex then moves to the nucleus, where it binds to κB enhancers in the regulatory regions of various genes, initiating transcription [104]. NF-κB target genes have diverse roles encompassing functions not limited to proliferation, survival, or angiogenesis [105]. 

With regard to NF-κB’s effect on cell proliferation and survival in lung cancer, NF-κB signaling activates cyclin D1 by binding to its promoter, promoting cell proliferation [97]. Cyclin-dependent kinases (CDKs) are key in regulating the cell cycle, as they interact with cyclin proteins [106]. Cyclin D1 binds to CDK6 and CDK4, causing Rb protein phosphorylation. This prevents Rb from inhibiting the E2F family transcription factors, facilitating the transcription of numerous genes required for the G1-to-S phase transition, and ultimately promotes cellular proliferation [107].

The Bcl-2 protein family, a group of proto-oncogenes, negatively regulate apoptosis and are often dysfunctional in various cancer types [98]. The human Bcl-2 promoter and inhibitors of apoptosis (IAPs) have been found to contain an NF-κB binding site, providing resistance to apoptosis induced by TNF-α [108]. Consequently, NF-κB activation in cancer cells during chemotherapy or radiation therapy is primarily associated with apoptosis resistance, a significant obstacle to effective cancer treatment [109]. In cancer cells, the NF-κB signaling pathway promotes angiogenesis, which involves regulating key pro-angiogenic factors such as VEGF and the proinflammatory cytokine IL-8 [99]. 

The STAT proteins regulate many aspects of growth, survival, and differentiation in cells. Ref. [110] STAT3 signaling is frequently overactive in most human cancers and serves as a recognized intrinsic pathway that promotes inflammation, cellular transformation, survival, proliferation, invasion, angiogenesis, metastasis, and immune evasion in cancer [100,101,111]. In addition, IL-6 activates STAT3 and is associated with advanced-stage disease and reduced survival in cancer [112]. There are also collaboration and crosstalk that occur between NF-κB and STAT3 in cancer progression, and efforts to target these pathways have yielded promising outcomes in cancer treatment [100,110,113].

The STAT proteins regulate many aspects of growth, survival, and differentiation in cells. Ref. [110] STAT3 signaling is frequently overactive in most human cancers and serves as a recognized intrinsic pathway that promotes inflammation, cellular transformation, survival, proliferation, invasion, angiogenesis, metastasis, and immune evasion in cancer [100,101,111]. In addition, IL-6 activates STAT3 and is associated with advanced-stage disease and reduced survival in cancer [112]. There are also collaboration and crosstalk that occur between NF-κB and STAT3 in cancer progression, and efforts to target these pathways have yielded promising outcomes in cancer treatment [100,110,113].

The activation of STAT3 in macrophages and neutrophils is essential for safeguarding against chronic inflammation [114]. Mutant mice lacking this activation exhibited heightened susceptibility to endotoxin shock, resulting in elevated levels of inflammatory cytokines, including TNF-α, IL-6, IL-1β, and IFN-δ in their serum [110]. 

## 4. Immunotherapeutic Strategies

Lung cancer can be subdivided into certain types. The most common division seen in the literature is NSCLC and SCLC. NSCLC has been found to have several common oncogenes, such as KRAS, c-MET, ALK, RET, BRAF, ROS1, NTRK, TP53, and ERBB2. Certain genomic alterations will allow for T cell-specific adaptive immune responses. The genomic alterations will cause a change in the APC-mediated antigen presentation. Immunotherapies focus on assisting the immune system in fighting the cancer cells. An important aspect of immunotherapies relating to cancer is the TME, which features a combination of ECM, fibroblasts, mesenchymal cells, and immune cells, among others.

### 4.1. Immune Checkpoint Inhibitors

TMEs contain immune checkpoints, which are receptor/ligand interactions that suppress the T cell response. Immune checkpoints are generally found in healthy cells; however, certain tumors express the checkpoint, preventing a T cell-mediated response. An important therapy that has emerged for NSCLC is the immune checkpoint inhibitor, which would stop the interaction of the T cells with immune checkpoints, including CTLA-4 and PD-1/PDL-1 [115]. Multiple studies have shown the therapeutic value of anti-CTLA-4, anti-PD-1/L1 and their combination, with or without chemotherapy in metastatic or advanced NSCLC (Table 2) [116,117,118,119,120,121,122,123].

### 4.2. CAR T Cell Therapy

Chimeric Antigen Receptors (CARs) are synthetic receptors that are transduced onto T cells. These receptors have activation sites that allow for increased T cell responses, often with costimulatory binding sites to modulate responses. EGFR-targeting CAR T cells are undergoing multiple clinical trials. In an ongoing phase 1 trial (NCT0415379; Table 2), 11 patients with EGFR-positive tumors are being administered anti-EGFR-modified CAR T cells. In a recent phase 2 trial (NCT01869166), patients with advanced NSCLC with an over 50% EGFR expression received the anti-EFGR CAR T cell treatment, and the patients were able to tolerate the therapy without severe toxicity for 3–5 days at a time [124]. Similarly, other tumor-specific molecules can be primed to the CAR T cells, allowing for patient-specific treatment. Further examples of such molecules include CEA, MUC1, MSLN, PD-L1, ROR1, and HER2 [124].

### 4.3. CAFs, TAMs, and TANs

CAFs, TAMs, and TANs all play important roles in tumorigenesis and subsequent immunotherapy. TAM will secrete cytokines that are involved in angiogenesis and tumor invasion along with immunosuppression [125]. CAFs have been shown to suppress the immune system through CTLA4 upregulation, inhibiting CD8+ T cells [126]. TANs have been found to secrete PGE2, which promotes NSCLC cell proliferation, causing increased tumor growth. Another important aspect of neutrophil involvement is ROS damage causing pre-disposed cells to go through an oncogenic transformation [127].

Several immunotherapeutic strategies have been proposed to use these cells as a means of curtailing tumor growth. IFN therapies have shown an increase in neutrophil antitumor therapy by way of ICAM1 upregulation. Increasing ICAM1 would increase immune activity [128]. A phase 1 clinical trial (NCT02001974) has shown the benefit of inhibiting the CXCR-1 and CXCR-2 chemokines in patients with triple-negative breast cancer using Reparixin in tandem with paclitaxel [129]. These chemokines recruit neutrophils to the tumor, creating the TAN and its subsequent protumor effects. An ongoing phase 2 study (NCT02370238) is evaluating the progression-free survival (PFS) of patients with TNBC treated with Repaxirin. There are currently no ongoing trials on anti-TAN therapy for patients with lung cancer. However, there are promising RNA vaccine therapies being explored that can use TAN-specific molecules as targets [130].

Some of the approaches taken to use CAF inhibition to curtail cancer growth include using CTLA-4 antibodies to block the effects of CAF and using NOX4 inhibition to prevent myofibroblast activation by inhibiting TGF-β activity. A phase 3 trial that evaluated antibodies that targeted TGF-β and PD-L1 in lung cancer was discontinued. Another important approach is the inhibition of FAP, or a fibroblast-activating protein using primed CAR T cells. A recent phase 1 clinical trial (NCT01722149) evaluated four patients with metastatic pleural mesothelioma and found that intrapleural injections of FAP CAR T cells increases proinflammatory cytokines in the sera with minimal upper respiratory infections and thromboembolic events [131].

TGF-β is involved in the differentiation of CAF, and Galunisertib is a TGF-β inhibitor [131]. A phase 1b/2 clinical trial (NCT02423343) featured 41 participants with advanced solid tumors and recurrent NSCLC or HCC. Researchers found that the MTD of galunisertib was 300 mg in phase 1b. In phase 2, the results showed that no patients were found to have anti-nivolumab antibodies after being administered galunisertib. The ORR was 24% in NSCLC patients, and the median PFS was found to be 5.26 months [132].

Several therapies target TAMs, such as sitravatinib, cabozantinib, bemcentinib, BA30011, and INCB081776 [133]. Sitravitinib targets receptor tyrosine kinases, including those that are found on the TAMs. In a phase 1 trial, sitravitinib was found to shift TAMs to an immunostimulatory state, increasing the ratio of M1 to M2 macrophages found in the TME [134]. In a phase 2 study on the same drug along with nivolumab in NSCLC patients with prior CPI therapy, the drugs were found to be clinically active together [135]. However, in the subsequent phase 3 trial, SAPPHIRE (NCT03906071), the combination of sitravatinib plus nivolumab did not improve survival when compared to docetaxel in patients with previously treated advanced nonsquamous NSCLC [136].

Cabozantinib is also a tyrosine kinase inhibitor that has been implicated in various cancers such as medullary thyroid cancer, hepatocellular carcinoma, and RCC [133]. The phase 3 CONTACT-01 trial compared cabozantinib with atezolizumab against docetaxel in NSCLC patients who had received platinum-based chemotherapy and CPI (NCT04471428) [133,137]. However, the study did not meet its primary endpoint of overall survival at the final analysis [138].

### 4.4. Oncolytic Viruses

Genetically modified viruses can be used for oncolytic purposes, resulting in inhibitory effects on tumor development [139]. The viruses will have a tropism for specific targets within cancer cells such as PSA or COX2, or even surface markers such as EGFR and CD20 [140]. There have been few clinical trials conducted on human subjects. The lysogenic adenovirus was found to result in extended disease progression in a phase 2 study (NCT01574729) that looked at rAd-p53 gene therapy combined with surgery on NSCLC. The study comprised 58 patients in stage III or IV NSCLC who received either a combination of the virus targeting the p53 gene (intervention arm) and chemotherapy through the bronchial artery or chemotherapy alone (control arm). The intervention arm was found to have an extended disease progression period, with an MS of 7.7 months as opposed to 5.5 months, *p =* 0.018. Two of the patients exhibited complete responses to the combinatorial treatment, both of whom had stage III NSCLC [141].

### 4.5. Tumor-Infiltrating Lymphocytes (TILs)

TILs are lymphocytes that are combined with specific T cell clones of tumor antigens, allowing for specialized tropism to the tumor. The approach consists of taking lymphocytes from the tumor and causing ex vivo proliferation using IL2 [142]. There have been several studies conducted on TILs in patients with lung cancer. An early study on TILs was performed on 131 stage II and III patients that had undergone a resection of NSCLC [143]. Ex vivo recombinant IL-2 was used to proliferate the extracted lymphocytes. The intervention arm of the study was found to have a higher MS than the control arm, 22.4 months as opposed to 14.1 months. A 3-year survival was found to be better for patients who underwent the TIL therapy (*p* < 0.05). TIL therapy has also been found to be effective in PD-1-resistant lung tumors in a phase 1 clinical trial (NCT03215810). A total of 20 patients with advanced NSCLC were administered autologous TIL along with nivolumab, and 11 patients were found to have a reduction in tumor burden (NCT03215810) [144].

### 4.6. IL-1β

IL-1β has been found to increase metastasis in lung cancer through angiogenesis, tumor epithelial-to-mesenchymal transition, adhesion, growth invasion, and cytokine production [3,145,146,147]. IL-1β has also been found to polarize M2 macrophages, increasing immune suppression and angiogenesis [148,149]. Currently, the major clinical program looking at IL-1β therapy in lung cancer is CANOPY (Canakinumab Outcomes in Patients with NSCLC Study). Canakinumbad is an FDA-approved human monoclonal antibody that targets IL-1β for acute systematic juvenile arthritis and periodic fevers [3]. The CANOPY program consists of six different trials. Half of the trials study the combination of anti-PD-1 and anti-IL-1β, which has shown an increase in CD8+ cytotoxic T cell infiltrate in the tumor [84,150]. Although the currently completed CANOPY trials did not yield positive results in the primary endpoints, they did demonstrate significant improvement in patient biomarker-based subgroups [3].

### 4.7. NF-kB

One of the major therapies that have been approved by the FDA that specifically targets VEGF by way of NF-kB is bevacizumab [102]. In the E4599 phase 3 trial, 15 mg/kg of bevacizumab in addition to carboplatin/paclitaxel was found to improve the median OS in patients with NSCLC, as opposed to chemotherapy alone. The trial yielded a hazard ration of 0.79 (*p* = 0.003), and a median OS of 12.3 months as opposed to 10.3 months with chemotherapy alone [151]. In the AVAiL trial, another large stage III randomized study, bevacizumab at 7.5 and 15 mg/kg was evaluated in addition to cisplatin/gemcitabine against chemotherapy alone in patients with NSCLC. The trial was found to increase PFS from 6.1 to 6.7 months in the 7.5 mg/kg dosage group (HR 0.75, *p* = 0.003) and to 6.5 months in the 15 mg/kg group (HR = 0.82, *p* = 0.03) [152]. Another important therapy that targets NF-kB is bortezomib. A phase 2 clinical trial (NCT00075751) examined the effects of bortezomib along with gemcitabine/carboplatin together in patients with stage IIIB/IV NSCLC. The study found that that median OS was 11 months, (95% CI: 8.2–13.4 months), and the median PFS was 5 months (95% CI: 3.5–5.3 months). Out of the 113 patients evaluated for safety, 3 patients had grade 3 hemorrhages, and 1 patient had febrile neutropenia [153].

### 4.8. IL-6

There are two current trials that are evaluating the efficacy of tocilizumab in lung cancer. Tocilizumab, an anti-IL-6 receptor antibody, has been found to improve cachexia in lung cancers that overexpress Il-6 in mice [154]. The first trial, NCT04940299, is a phase 2 clinical trial that is studying the effects of tocilizumab, nivolumab, and ipilimumab in patients with NSCLC, urothelial carcinoma, and melanoma. The other trial, NCT04691817, is a phase 1/2 trial examining tocilizumab in combination with atezolizumab in patients with NSCLC that is either locally advanced or that has metastasized and has not responded to treatment (Table 2).

### 4.9. STAT3

A phase 1 trial of OPB-51602 (NCT01184807) was found to achieve partial responses in two patients with NSCLC [155]. However, there are no substantive human trials currently that have assessed the relationship between STAT3 inhibition and lung cancer. A promising trial that is currently underway is a phase 2 trial looking at danvatirsen and durvalumab in patients with advanced and refractory pancreatic cancer, NSCLC, and colorectal cancer. Danvatirsen (AZD9150) is an inhibitor of STAT3 (NCT02983578).

### 4.10. TNF-α

Certozilumab is an important inhibitor of TNF-α that underwent a phase 1 trial for patients with stage IV lung adenocarcinoma. The study involved chemotherapy combined with certolizumab to evaluate the toxic effects of the drug (NCT02120807, Table 2). The trial found that the standard dose of 400 mg of certolizumab was well tolerated and had potential for further study [156].

### 4.11. IL-8

There is one ongoing trial that is examining the effect of administering nivolumab with anti-IL-8 versus a CCR2/5 inhibitor both before surgery and after surgery in patients with NSCLC or hepatocellular carcinoma (NCT04123379, Table 2).

### 4.12. IL-10

Pegilodecakin, or recombinant IL-10, has undergone two major phase 2 randomized controlled trials to test its effectiveness in relation to NSCLC. In CYPRESS 1 (NCT03382899, Table 2), the control arms were administered pembrolizumab, and in CYPRESS 2 (NCT03382912, Table 2), the control arms were administered nivolumab, while the experimental arms were administered pegilodecakin with the respective checkpoint inhibitors. CYPRESS 1 featured a median PFS increase from 6.1 to 6.3 (hazard ratio of 0.937), and a median OS of 16.3 months as opposed to not reached (hazard ratio = 1.507). CYPRESS 2, however, had a median PFS of 1.9 in both arms, and a median OS of 6.7 months in the experimental arm vs. 10.7 in the control arm [157]. 

## 5. Challenges and Future Directions

Despite the various mechanisms of immunotherapy at play in trials on lung cancer, there remain many challenges to address on the road to the overwhelming goal of curing cancer. In particular, the patient-specific differences in response to the same drug create a lack of efficacy, as well as the various resistance mechanisms at play and the difficulty in bringing scientific success into the clinic. Going forward, these challenges need to be addressed across all fronts. An expansion of known biomarkers will help understand case-by-case variations in responses, while a stronger understanding of the immune system will open the door to how it can best be utilized and modified. Lastly, a stronger focus on combinational therapy and more improved treatment directives will help create more durable clinical successes and improve the standardization of future research.

### 5.1. Case-Specific Variations in Efficacy

One of the most pressing challenges in the treatment of many cancer types, particularly lung cancers, is that the current approaches developed have not been able to yield a broadly applicable treatment. Within this challenge exists various proposed causes: variations, previous treatment history, inherent immunosuppressive features of malignancy, type and stage of growth, TME heterogeneity, and pathways for malignancy [158].

Because of how specific current research is on the direct inhibition of single molecules, the efficacy of these treatments is limited to a small population, and even within that population, there is no guarantee of success. As mentioned, ICBs are common tools in the repertoire of oncologists treating cancer; however, their success has not been universal, given the heterogeneity of immune regulation systems and the difficulty in targeting them [115,159]. Their lack of use as first-line therapy in many cancers also represents difficulties when discussing their efficacy as they are commonly administered after chemotherapy in patients with impaired immune responses [158,159]. Moreover, as the literature continues to shift toward the discussion of various combinational therapies, it will become increasingly difficult to identify which of these is the most recommended for a particular patient. The validity of immunotherapy is quite strong, particularly in NSCLC with brain metastases, where immune checkpoint inhibitor treatments produced longer PFSs and OSs, and these results are improved only when combined with chemotherapy [160]. However, it is important to note that across the spectrum of lung cancer and particularly NSCLC, the number of patients qualifying for each therapy or responding strongly to it is highly variable.

### 5.2. Resistance Mechanisms

Many of the mechanisms discussed as part of proliferative or survival pathways are potentially active in resistance against immunotherapeutics. For example, the creation of subclonal tumor cells that do not express neoantigens dampens the possible response to immunotherapies dependent on T cell cytotoxicity [161]. Further, heterogeneity in the mechanisms responsible for growth, antigens expressed, deficiency in antigen presentation, low tumor mutation burden (TMB), and PD-L1 expression are all possible mechanisms of intrinsic resistance. As identified in Figure 1, there are also various TME-related immune modulations that can all pose possible extrinsic resistance mechanisms. For example, host immunosuppressive cells being recruited or activated, T cell exhaustion, cytokine or chemokine alterations, and increases in immune surveillance avoidance mechanisms (i.e., increased PD-1 expression) all contribute to the heterogeneity of resistance against immunotherapy [157,161]. Along these lines, SCLC has been particularly difficult to treat with immunotherapy. Despite data from clinical trials of checkpoint inhibitors providing hints at improved long-term survival, the immunosuppressive TME, with low PD-L1 and MHC antigen expressions as well as avascularity that restricts immune reach, has created an environment resistant to immune modulation [162,163]. Trying to understand these changes and interactions should be at the forefront of the oncological approach, as without an adequate understanding of these mechanisms, the same pitfalls against treatment will continue to dampen the efficacy of immunotherapy.

### 5.3. Bringing Scientific Success into the Clinic Going Forward

Currently, the PD-L1 expression is the most widely used biomarker in the clinic; however, its value as a predictive marker suffers at the hands of tumor heterogeneity as well as differences in defining those variable expression levels [162,164]. Several potential biomarkers in various stages of development propose a possible answer to this issue. As possible alternatives to the standard immunohistochemical methods of analyzing PD-L1 expression, turning more toward gene expression-based, TMB, CBC, peripheral blood mononuclear cell (PBMC), TIL, extracellular vesicle, imaging, and microbiome biomarkers may provide a more complete picture of the tumor and its relationship in immunotherapeutic treatment. 

Another promising approach that is now at the forefront of many clinical trials and studies includes combinations of immunotherapy with a plethora of other treatment types. For example, the double barricade approach of combining checkpoint inhibitors like anti-PD-L1 and anti-CTLA4 demonstrates a potential future for immunotherapy as each inhibitor’s unique abilities to modulate the TME propose a more durable protection against TME-mediated resistance [165]. Other combinations like those with DNA repair targeting agents, targeted agents (i.e., G12C inhibitors, EGFR inhibitors, etc.), chemoradiation, chemotherapy, and novel checkpoint inhibitors that are currently undergoing trials also represent a new frontier in immunotherapeutics aiming for a more complete treatment approach [162]. In any of these approaches, there exists a need for improved clinical trial designs that take into account the likelihood of a delayed immune response, baseline immune statuses, and the difficulty in expanding qualified cohorts [159].

## 6. Conclusions

The inflammatory TME plays a central role in lung cancer development, fostering an environment conducive to oncogenesis while also shaping the efficacy of immunotherapies. Recent advances in our understanding of paracrine regulation, immune pathways, and molecular mechanisms offer valuable insights for devising novel strategies to combat lung cancer. Despite the current difficulties in maintaining efficacious treatments across a larger population and addressing the hypervariable resistance mechanisms that come with them, the future of immunotherapy is anything but bleak. By developing a more thorough understanding of the intricate interplay between inflammation, immune response, and the TME, we can potentially enhance the effectiveness of immunotherapeutic interventions, paving the way for improved outcomes in lung cancer treatment. 

## Figures and Tables

**Figure 1 cancers-16-01113-f001:**
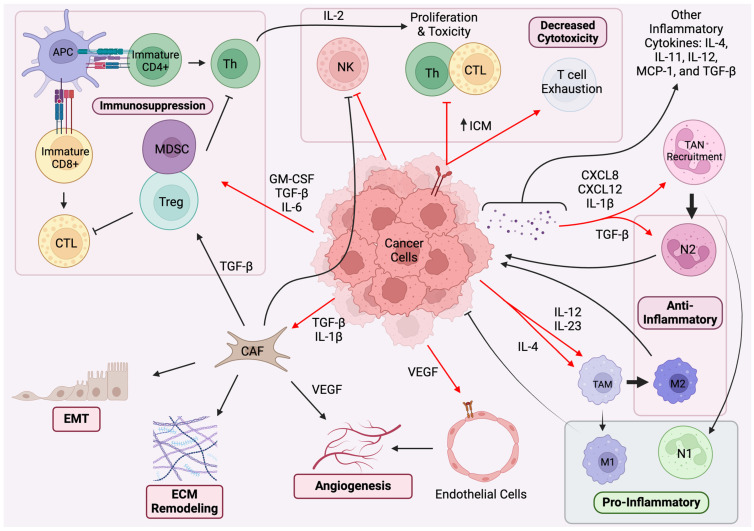
Paracrine regulation and immune system pathways in the TME. APCs (e.g., DCs, macrophages) sample antigens from cancerous cells and present the antigen on MHC class I to CD8+ T cells and MHC II class 2 to CD4+ T cells. After activation by the APC, CTLs and Th cells migrate to the site of inflammation and induce a cytotoxic response directly (CD4+ and CD8+) or indirectly through other effector cells (CD4+). CAFs in the TME stroma have various functions including the recruitment of immunosuppressive cells (e.g., Tregs and MDSCs), increasing the expression of immune checkpoint receptors, facilitating EMT, remodeling the ECM, and angiogenesis. The induction of immuno-suppressive cell phenotypes through cancer cell-mediated pathways results in a reduction in anti-inflammatory immune function. Cancer cells also secrete VEGF, which stimulates proliferative angiogenesis as it interacts with ECs. TAMs are distinguished into two main subclasses: M1 TAMs are involved in proinflammatory antitumor reactions through cytokines like IL-12, whereas M2, is associated with tissue repair and thus is anti-inflammatory through IL-10. M2 polarization through IL-4 commonly occurs in lung cancers and contributes to various mechanisms of tumor growth. Cancer cells themselves exhibit changes in expressions including an increase in ICMs like PD-L1 and a reduction in MHC class I expression, therefore diminishing CTL recognition. Chemokines are involved in immune cell trafficking within the TME and, specifically, with the recruitment of TANs. Subpopulations of TANs include the antitumor N1 and protumor N2, which is favored in a proliferative TME with the production of TGF-β. Several other cytokines and chemokines produced by cancer cells are involved in the inflammatory environment as well. Red arrows denote cancer cell function. Black arrows denote stromal cell functions. Arrowheads represent stimulation or increases, while flat-heads represent inhibition or reductions. (Created using BioRender.com). Abbreviations used: APC = antigen-presenting cell; Th = helper T cell; CTL = cytotoxic T lymphocyte; Treg = regulatory T cell; MDSC = myeloid-derived suppressor cell; NK = natural killer cell; ICM = immune checkpoint molecules; EMT = epithelial–mesenchymal transition; ECM = extracellular matrix; CAF = cancer-associated fibroblast; TAN = tumor-associated neutrophil; TAM = tumor-associated macrophage; IL-# = interleukin-#; TGF-β = transforming growth factor-beta; VEGF = vascular endothelial growth factor; GM-CSF = granulocyte macrophage colony-stimulating factor; CXCL = chemokine (C-X-C motif) ligand; MCP = monocyte chemotactic protein.

**Table 2 cancers-16-01113-t002:** Ongoing clinical trials exploring regulation of TME.

Target	Treatment	Mechanism	Phase	Clinical Trial ID
CTLA-4	Iplimumab	Immune Checkpoint Inhibitor	3	NCT00527735
PD-1	Nivolumab	Immune Checkpoint Inhibitor	2	NCT02998528
EGFR	N/A	CAR T Cell Therapy	2	NCT01869166
	N/A	CAR T Cell Therapy	1	NCT0415379
TGF-β	Galunisertib	CAF Inhibition	1b/2	NCT02423343
Receptor Tyrosine Kinase	Siravatinib, Nivolumab	TAM Inhibition	3	NCT03906071
Tyrosine Kinase	Cabozantinib, Atezolizumab	TAM Inhibition	3	NCT04471428
IL-1β	Canakinumbad, Pembrolizumab, Chemotherapy	IL-1β Inhibition	3	NCT03631199
	Canakinumbad, Chemotherapy		3	NCT03626545
	Canakinumbad, Pebrolizumab		2	NCT03968419
	Canakinumbad		3	NCT03447769
	Canakinumbad, PDR001		1b	NCT02900664
	Canakinumbad, PDR001+		1b	NCT03064854
P53 Gene	N/A	Oncolytic Virus	2	NCT01574729
N/A	N/A	TIL	3	N/A
N/A	N/A	TIL	1	NCT03215810
NF-kB	Bevacimuzab	VEGF Inhibition	3	NCT00021060
NF-kB	Bevacimuzab	VEGF Inhibition	3	NCT00806923
NF-kB	Bortezomib	NF-kB Inhibition	2	NCT00075751
IL-6	Tocilizumab	IL-6 Antibody	2	NCT04940299
IL-6	Tocilizumab	IL-6 Antibody	1/2	NCT04691817
STAT3	OPB-51602	STAT3 Inhibitor	1	NCT01184807)
STAT3	Danvatirsen, Durvulamab	STAT3 Inhibitor	2	NCT02983578
TNF-α	Certizolumab	TNF-α Inhibitor	1	NCT02120807
IL-8	BMS-986253	IL-8 inhibitor	2	NCT04123379
N/A	Pegilodecakin	Recombinant IL-10	2	NCT03382899
N/A	Pegilodecakin	Recombinant IL-10	2	NCT03382912

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
