# Peer review of "Paracrine Regulation and Immune System Pathways in the Inflammatory Tumor Microenvironment of Lung Cancer: Insights into Oncogenesis and Immunotherapeutic Strategies"

_cancers, 2024, doi:10.3390/cancers16061113_

Round 1
Reviewer 1 Report
Comments and Suggestions for Authors
In this review, authors have discussed the interplay of various components of the tumor microenvironment (TME) and their role in development of therapeutic strategies for lung cancer. However, the major concern is that authors have discussed how the interactions within the TME are critical for the lung cancer carcinogenesis- this is contradictory. The molecular mechanisms discussed are more relevant for cancer growth, progression, angiogenesis, survival, and metastasis rather than the process of carcinogenesis (initiation of tumor/cancer; before the tumor or TME is established). Therefore, the title as well as the text throughout the review (ex. lines 277, 615 etc.) should be changed accordingly.
Minor comments:
1. Line 25: Add cancer cells as part of the TME.
2. Line 83: The interaction is….….which- this part of the sentence is not required as this is already stated in the previous sentence.
3. Line 92: “The inflammatory cytokines, such as TNF-α and IL-6, are pivotal mediators of immune responses in the TME aiding in the anti-tumor defense mechanisms.” This is already stated in section 2.1.
4. Line 275: Rephrase the sentence.
5. Table 1: Include another column with corresponding references.
6. Table 2: is referred only in the context of CAR T cell therapy but this should be referred at other places, ex. for ICI, TAM inhibition etc.
7. Line 415: against docetaxel (not clear).
8. Line 436: TILS (change to TILs).
9. Table 2: Fix kappa in NF-kappa B (also in some other parts of the text).
10. Line 461: Correct to15mg/kg (from kb).
11. Line 492: Rephrase.
12. Line 493: “There is one ongoing trial that is examining the effect of the giving nivolumab…..” Change to “effect of giving nivolumab”.
13. Line 510: What is [Spigel]?
14. Line 550: Remove “previously”.
15. Line 599: Correct “an” to “in”.
Reviewer 2 Report
Comments and Suggestions for Authors
Overview
This paper thoroughly examines the impact of paracrine signaling and immune responses within lung cancer's tumor microenvironment. It provides comprehensive research on the dynamic interactions between tumor and immune cells, emphasizing the TME's complexity and its implications for cancer progression and treatment. The review also identifies promising therapeutic targets, suggesting directions for future research to enhance lung cancer therapies.
General comments and questions
There are quite a few abbreviations used in the manuscript, some of them are not well explained the first time it was introduced. Considering add such information. Also, considering adding a list of abbreviations for better readability.
Specific comments and questions
Line 46-48: It is not clear what is the relationship between CAFs and these C-X-C type chemokines. Please elaborate on whether CAFs directly secrete them. In additional, please double check the relationship of CAFs and CXCL13, considering adding this info if necessary.
Line 57-63: To provide a better characterization of endothelial cells, consider adding information about the interaction between endothelial cells and immune cells where paracrine regulation is involved.
In section 2.2: In order to provide a comprehensive overview of pathways in the immune system in the TME, considering adding more information by touching on pathways involving other relevant cells, for examples, MDSCs, NK Cells, Dendritic Cells.
Considering addressing the issue that the watermark is overlapping with texts in figure 1.
Reviewer 3 Report
Comments and Suggestions for Authors
This review article is focused on paracrine regulations mediating the interplay between inflammatory processes and the tumor microenvironment. Although this work is specifically dedicated to the lung cancer, the presented summarized data and conclusions are important for understanding of the mechanisms of carcinogenesis and mutual interactions between the TME components in general. Selected cytokines/chemokines, their origin in the TME, mechanisms of their action, including corresponding principle intracellular pathways’ descriptions are mentioned and discussed.
This review article is well-written and provides concise and up-to date information. The topic of the article is not unique as a number of similar articles summarizing regulations and interactions within the TME and already exists. Importantly, the chapter dedicated to the ongoing clinical trials exploring regulation of the TME brings a very useful piece of information interesting not only for experts in the field.
Specific comment:
The topic is very complexed and a single review cannot cover all players and interactions in the TME. However, interferons should not be omitted in this review as they are one of the principle regulators in the TME that can influence not only non-tumor cells but also directly proliferation an immune recognition of cancer cells. So, their roles should be discussed and included into the net of the paracrine regulators.
Reviewer 4 Report
Comments and Suggestions for Authors
Title: Paracrine Regulation and Immune System Pathways in the Inflammatory Tumor Microenvironment of Lung Cancer: Insights into Carcinogenesis and Immunotherapeutic Strategies
Summary: In this review, authors addressed the most current knowledge and recent advances on paracrine regulation and role of immune mechanism in lung cancer. Authors have discussed the current scientific knowledge, ongoing clinical updates, and future challenges from the perspective of TME and its role in shaping the tumor.
Major comments: None
Author Response
Thank you for your time reading. No revisions.